# Exploring Text-to-Motion Generation with Human Preference

## Abstract

*This paper presents an exploration of preference learning in text-to-motion generation. We find that current improvements in text-to-motion generation still rely on datasets requiring expert labelers with motion capture systems. Instead, learning from human preference data does not require motion capture systems; a labeler with no expertise simply compares two generated motions. This is particularly efficient because evaluating the model's output is easier than gathering the motion that performs a desired task (e.g. backflip). To pioneer the exploration of this paradigm, we annotate 3,528 preference pairs generated by MotionGPT, marking the first effort to investigate various algorithms for learning from preference data. In particular, our exploration highlights important design choices when using preference data. Additionally, our experimental results show that preference learning has the potential to greatly improve current text-to-motion generative models. Our code and dataset will be publicly available to further facilitate research in this area.*

## 1. Introduction

Human motion generation [2, 6, 12, 15, 19, 27, 30, 33, 44, 47–51] is a profoundly pertinent task with extensive applicability in computer animation, movie production, gaming, and robotics. However, current motion generation research relies on relatively modest datasets compared to language tasks, as expert labelers with specialized motion capture systems are costly and labor-intensive. Due to the lack of large-scale data, these models are poorly aligned with the text prompt [19, 47, 51].

Learning from preference data [26, 32, 54] has emerged as a powerful novel training paradigm in cases where evaluation proves simpler than generation. With a simple data collection pipeline where layman labelers compare two motion sequences, preference data gives us extremely cost-effective labels to improve motion generation models without expert labelers.

While learning from preference data has excelled in domains abundant with datasets, particularly in language tasks benefiting from ample and high-quality data, its application in fields constrained by limited, multi-modal data presents a unique challenge. The current landscape of learning from preference data is rife with intricate engineering details and subtle design choices, often concealed within implementations and validated solely through empirical experimentation [26, 36, 40, 53]. Yet, there are currently no existing motion datasets tailored for exploring preference learning techniques. As a result, initiatives to extend preference learning to these low-data, multi-modal setups remain absent, for they lack empirical evidence to substantiate the intricate design decisions pivotal for applying preference learning in motion generation. This absence underscores an intriguing gap in our understanding and presents an exciting opportunity to investigate how preference learning performs in motion generation tasks where data is scarce.

Previous endeavors address data scarcity by aligning to large language models' (LLMs) rich representation [19, 51]. While this approach transfers some of the compositional structure of language, it nonetheless requires a large dataset of text and motion pairs, failing to circumvent the data issue. Other methods resort to pseudo-labeled data [21] to offset the dearth of large-scale datasets in motion generation. However, such approaches often introduce noisy learning signals that may amplify problems, such as perceptually unrealistic motions. Alternatives involve injecting noise into existing labels and learning from the resulting ranked generations [41]. Nonetheless, this approach neglects training on the actual policy distribution, leading to a distributional gap wherein the reward model is not trained to supervise the actual policy.

We classify existing methodologies according to the approximations employed to represent the preference distribution. In particular, current methods make one or both of the following approximations. First, they assume that pairwise preferences can be substituted by a scalar reward. In particular, they employ the Bradley-Terry probabilistic model [5] to connect scalar rewards to preferences. Second, they assume that a reward model trained with the Bradley-Terry model generalizes so that it can accurately evaluate sam-

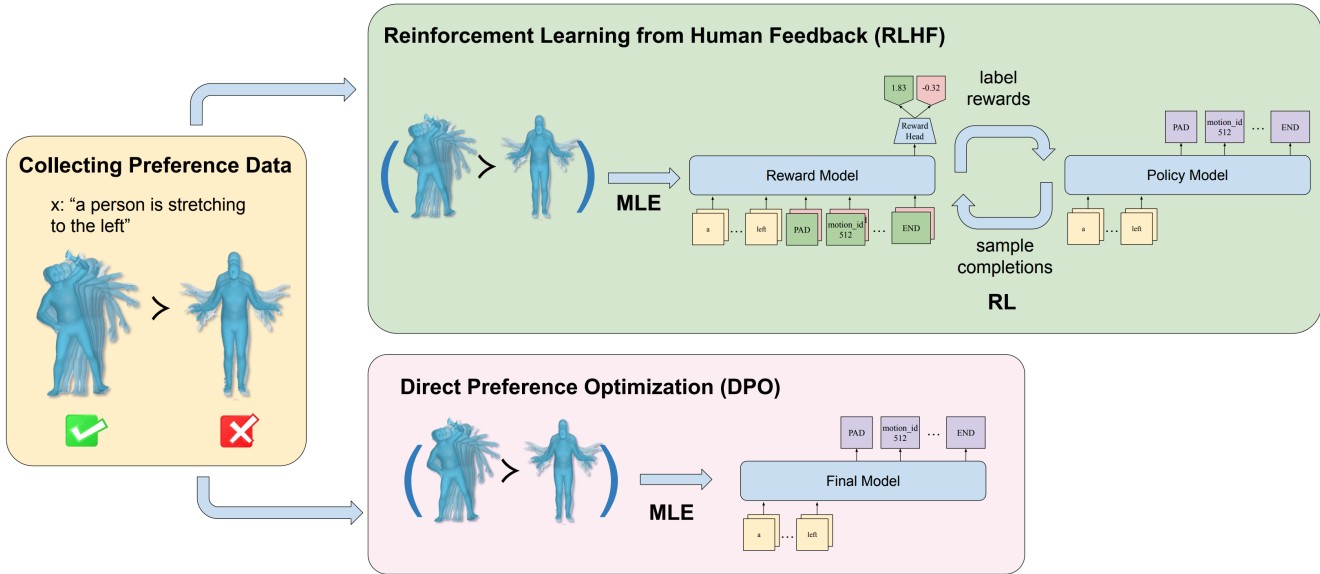

Figure 1. **Text-to-Motion Generation with Human Preference.** We gather preferences over generated completion (*i.e.*, motion) pairs and use them to finetune MotionGPT. In preference learning, the likelihood of preferred completion is increased while that of dispreferred completion is decreased. We explore two types of practical algorithms for preference learning. First, RLHF trains in an online manner; it trains a reward model on the data and uses it to perform RL on MotionGPT. Second, DPO trains in an offline manner with supervised learning; it directly performs MLE on the data. The online/offline aspect is related to whether or not the policy performs exploration, *i.e.*, training on completions outside of the preference dataset.

ples from the policy. Notably, it uses reinforcement learning (RL) to finetune against the reward model. While reinforcement learning from human feedback (RLHF) makes both assumptions, direct preference optimization (DPO) bypasses the RL step. RLHF [26] trains a reward model in a supervised way on the preference data, then finetunes the policy by optimizing against that reward model using reinforcement learning. In contrast, DPO [28] directly finetunes the preference data in a supervised manner using cross-entropy. DPO is a simpler algorithm, yet it lacks a crucial element found in online RL-based algorithms: exploration. By training a reward model, we can generalize to unseen samples. Accordingly, our policy can generate samples outside of the preference dataset (*i.e.*, exploration). In other words, RLHF allows us to get a training signal where the reward model generalizes via trial and error, thereby acquiring more information. Conversely, DPO is limited to two points within the data, optimizing to maximize one while minimizing the other.

We explore the aforementioned methods along their variants, and summarize our contribution as follows:

1. We annotate $3,528$ preference pairs generated by MotionGPT [19]. Additionally, we provide a degree of preference for each choice.
2. We are the first to demonstrate effective implementation of preference learning on motion generation models. Our results show that labelers exhibit a significant preference for outputs from MotionGPT when trained with preference data, a trend that persists across temperatures ranging from 1.0 to 2.0.
3. Our findings indicate that the scarcity of large-scale text-motion pairs leads to a propensity for the reward model to overfit. Consequently, this overfitting hampers its ability to accurately assess outputs generated by MotionGPT. In light of this, we propose the adoption of DPO, a method that circumvents the optimization over a reward model, thereby avoiding reward hacking.
4. We find that labels characterized by a pronounced degree of preference significantly contribute to the observed enhancement in R-precision. This suggests that the differential quality of preference annotations plays a pivotal role in driving the efficacy of the model.

We organize this paper as follows. We present related works in Sec. 2. Our work finetunes upon MotionGPT [19], which we present in Sec. 3. We detail the implementation of our data collection pipeline alongside specific design details for RLHF and DPO in Sec. 4. Experimental results in Sec. 5 illustrate our key design choices. Finally, Sec. 6 contains a summary of our findings and discusses future work.

## 2. Related Works

### 2.1. Autoregressive Motion Generation

Numerous motion generation methods leverage diffusion models to generate motion sequences [2, 6, 27, 30, 33, 44, 48–50]. However, human motion inherently exhibits semantic connections and is frequently interpreted as a form of body language, conveying meaning and intent. Following this observation, several works have explored treating motion as a form of language and using the generative transformer framework to model human motion, akin to the current methods for modeling language [19, 51]. This approach involves converting motions into discrete tokens using vector quantization (VQ) [37] and inputting them into an autoregressive model to generate a sequence of motion tokens in a unidirectional manner [12, 15, 47]. Subsequent works also leverage pretrained LLMs such as T5 [29] and LLaMA [35] to conduct comprehensive language modeling on both textual and motion inputs by expanding the existing LLM vocabulary with motion tokens [19, 51]. In this work, we build upon autoregressive Transformers [38], which have tractable log-likelihood, an essential element for preference learning methods.

### 2.2. Learning from Human Preferences

The initial exploration of learning from human preferences begins in the RL community with training agents to play Atari [8, 17]. Further exploration occurs in the domain of language modeling, where human feedback is incorporated to improve specific tasks like summarization [32, 54] and using external information to increase accuracy [23, 24, 34].

Building upon the aforementioned works, Ouyang et al. [26] shows that a blend of instruction fine-tuning and RLHF effectively addresses issues related to factuality, toxicity, and helpfulness, which cannot be resolved solely by increasing the scale of LLMs. Leveraging the proposed RLHF framework, numerous LLMs [11, 25, 36] incorporate the RLHF phase into their training process to mitigate potential model-related harm. The research community is also increasingly exploring other human preference learning methods [4, 7, 10, 13, 28, 31, 43, 52] that mitigate certain issues associated with RLHF, such as reward hacking [28], requirements for preference pairs [10], and complex hyperparameter tuning [43].

Motivated by the very successful application of preference learning in language modeling, preference learning is now increasingly being applied to other domains. For example, Lee et al. [20] and Wu et al. [42] apply RLHF to text-to-image synthesis models, and Cideron et al. [9] utilizes RLHF for music generation. Despite its promising potential, the research community has yet to witness its application in motion generation or scenarios with limited data resources. This untapped area of exploration represents a significant opportunity to advance our understanding of how these methods can be leveraged effectively in contexts where data availability is constrained, thereby opening new avenues for research and innovation in motion generation and related disciplines. Its application is particularly relevant to motion generation, where evaluating two motions is considerably easier than collecting motion data with costly motion capture systems.

## 3. Preliminary

This section reviews MotionGPT [19], the supervised baseline upon which we use preference learning to finetune. Additionally, we present our data collection pipeline that uses sample pairs generated by MotionGPT.

**MotionGPT.** Formulating text-to-motion generation as a sequence modeling problem allows building upon LLMs. This holds the premise of transferring language's compositional semantic structure to other modalities, thereby achieving off-the-shelf, out-of-distribution generalization. Casting motion generation as a sequence modeling problem requires discretizing the motion modality into tokens, as done by MotionGPT. The discretization process is akin to the tokenization of strings to tokens in language processing. In particular, they first map the motion dataset into a set of discrete tokens using a vector quantized variational autoencoder [37]. Then, a pretrained LLM is finetuned to generate corresponding motion tokens from the textual prompt. Thus, MotionGPT is an autoregressive model where the completions are motion tokens instead of word tokens.

**Collecting preference data.** As shown in Fig. 2, we build a labeling platform with Gradio [1], where labelers are presented with two different completions from a prompt. The labelers are tasked to read each prompt and choose the motion that corresponds best to the prompt. Additionally, the labelers provide a degree of preference for their choice, choosing from "Negligibly better/unsure," "Slightly better," "Better," and "Much better." We find that MotionGPT produces samples that are hard to distinguish when given a prompt from the training dataset, thus indicating signs of overfitting. Accordingly, we prompt gpt-3.5-turbo-0125 [25] to generate a new set of prompts similar to those in the training set. For each prompt, we sample two completions from MotionGPT by using different seeds and a temperature of 1.2 to promote diversity. Our labelers are graduate student in computer science. We find that it is important to recruit labelers with prior exposure to generative models. Our initial exploration indicates that labelers with similar prior experience is crucial for achieving a high level of agreement. Quantitatively, we obtain an agreement of $84\%$ on average (42 samples out of 50 samples). Note that in some cases, the model completely fails to generate perceptually realistic motion for both seeds. Accordingly, the labelers mark them as "Skipped." Upon inspection, we find

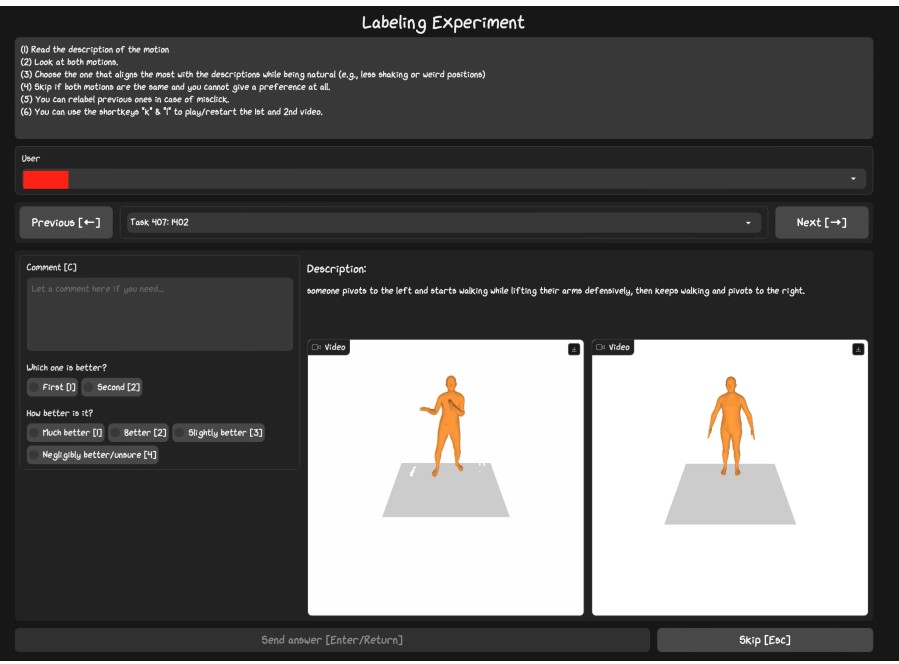

Figure 2. **Screenshot of the Gradio interface for data labeling.**

that two cases often occur: (1) one generation is "Much better," with one seed failing to generate reasonable motion while the other seed partially achieving the motion, (2) the motion pair is "Skipped," with both seeds failing to generate reasonable motions. The resulting dataset contains $3,528$ annotated pairs, with 996 pairs labeled as "Much better," 607 pairs labeled as "Better," 497 pairs labeled as "Slightly better," and 116 pairs labeled as "Negligibly better/unsure." Additionally, there are 1312 examples labeled as "Skipped." We randomly select 10% of the total dataset to be the test dataset, with the remaining data designated for training.

## 4. Method

We organize this section as follows. Sec. 4.1 presents the objective function in preference learning. Then, we present practical algorithms for optimizing this objective: RLHF based on reinforcement learning in Sec. 4.2 and DPO based on supervised learning in Sec. 4.3.

**Notations.** Denote sequences of tokens in bold where $\mathbf{x} = (x_1, x_2, ...)$ is a textual prompt and $\mathbf{y} = (y_1, y_2, ...)$ is a completion, *i.e.* a generated motion sequence.

### 4.1. Preference Learning

We formulate the objective function for learning from human preference data as in Azar et al. [3]. Intuitively, given prompts $\mathbf{x} \sim \rho$, it involves maximizing the probability that our policy generates completion $\mathbf{y} \sim \pi_\theta(\cdot \mid \mathbf{x})$ preferred over the original model $\mathbf{y}' \sim \mu(\cdot \mid \mathbf{x})$, under the constraint that our distribution stays close to that of some reference

policy $\pi_{\text{ref}}$ to prevent over-optimization. In most cases, $\mu$ and $\pi_{\text{ref}}$ are the same model, but it is not uncommon to initialize them differently. In formulae, we maximize the following objective in preference learning:

$$J(\theta) = \mathbb{E}_{\substack{\mathbf{x} \sim \rho \\ \mathbf{y} \sim \pi_\theta(\cdot|\mathbf{x}) \\ \mathbf{y}' \sim \mu(\cdot|\mathbf{x})}} \left[ \Psi(p^\star(\mathbf{y} \succ \mathbf{y}' \mid \mathbf{x})) \right] - \beta \mathbb{KL}(\pi_\theta \| \pi_{\text{ref}}),$$

(1)

where $p^\star(\mathbf{y} \succ \mathbf{y}' \mid \mathbf{x})$ is the probability of $\mathbf{y}$ being preferred to $\mathbf{y}'$ knowing the prompt $\mathbf{x}$. First, $\Psi : [0, 1] \rightarrow \mathbb{R}$ is a non-decreasing function that maps probabilities to real scalars. Intuitively, such mapping allows a non-linear mapping of preference probabilities to scores, yielding a reward maximization objective. Second, the KL term is a *per-token* KL that regularizes training in two ways. In formulae, the KL term can be rewritten as

$$\mathbb{KL}(\pi_\theta \| \pi_{\text{ref}}) = \underbrace{\mathbb{H}[\pi_\theta, \pi_{\text{ref}}]}_{\text{cross entropy}} - \underbrace{\mathbb{H}[\pi_\theta]}_{\text{entropy}}.$$

(2)

The cross-entropy term acts as a regularizer that prevents deviating too far from the reference model. It helps against hacking the objective function. The entropy term promotes exploration. It prevents the model from mode collapse, where the policy outputs sequences with high scores but low diversity. We want to maximize the score while maintaining a low KL divergence with high entropy.

As illustrated in Fig. 1, there are two types of algorithms for solving the optimization problem in Eq. 1. In both al-

gorithms, the underlying assumption is that the probability $p^\star(\mathbf{y} \succ \mathbf{y}' \mid \mathbf{x})$ is implemented as the Bradley-Terry probabilistic model [5]. Accordingly, we have $\Psi(q) = \log(q/(1-q))$. In practice, the Bradley-Terry model is implemented as a sigmoid function $\sigma$:

$$p^\star(\mathbf{y} \succ \mathbf{y}' \mid \mathbf{x}) = \sigma(r(\mathbf{x}, \mathbf{y}) - r(\mathbf{x}, \mathbf{y}')), \qquad (3)$$

thus the probability of preferring a completion depends *exponentially* on the value of a latent scalar.

Next, to understand the difficulties induced by the Bradley-Terry model in Eq. 3, we turn to the analytical optimal solution to the objective in Eq. 1:

$$\pi_{\theta^\star}(\mathbf{y} \mid \mathbf{x}) = \frac{1}{Z(x)} \pi_{\text{ref}}(\mathbf{y} \mid \mathbf{x}) \exp\left(\beta^{-1} r(\mathbf{x}, \mathbf{y})\right). \qquad (4)$$

As detailed in Eq. 4, the probability we assign to a particular response is the product of the probability that our reference model assigns to that response and the exponentiated latent scalar. The problem is that if $p^\star(\mathbf{y} \succ \mathbf{y}' \mid \mathbf{x}) = 1$, it means that $r(\mathbf{x}, \mathbf{y}) \to \infty$. As a result, the strength of the KL divergence $\beta$ vanishes, and the model is *prone to overfitting*.

We just observed that the current implementation of $\Psi$ assumes that pairwise preferences can be substituted with pointwise rewards. Next, we present RLHF in Sec. 4.2 and DPO in Sec. 4.3, the two most commonly taken approaches in LLM alignment. Notably, these two algorithms differ in being online or offline. RLHF is online because it trains with RL, *i.e.* there is exploration. At each step, a policy generates samples and receives feedback from a reward model. In particular, it assumes that a reward model trained on pointwise rewards generalizes so that it can accurately evaluate samples from the policy. DPO, on the other hand, is offline because it operates without the continuous interaction with the environment; instead, it optimizes based on predetermined data points.

### 4.2. RL with Human Feedback

RLHF is a bi-level optimization problem involving learning a reward model $r_\psi(\mathbf{x}, \mathbf{y})$ in a supervised manner, with a cross-entropy loss between the distribution of preference and the Bradley-Terry model. Given a dataset of preferences $\mathcal{D} = \{\mathbf{x}, \mathbf{y}_w, \mathbf{y}_l\}$, where $\mathbf{y}_w$ is the chosen sample, $\mathbf{y}_l$ is the rejected sample, and $\mathbf{x}$ is the input prompt, we minimize the following cross-entropy loss:

$$-\mathbb{E}_{(\mathbf{x}, \mathbf{y}_w, \mathbf{y}_l) \sim \mathcal{D}}[\log \sigma(r_\psi(\mathbf{x}, \mathbf{y}_w) - r_\psi(\mathbf{x}, \mathbf{y}_l))]. \qquad (5)$$

Then, we define Eq. 1 in terms of the trained reward model $r_\psi(\mathbf{x}, \mathbf{y})$ as an approximation of $\Psi(\cdot)$:

$$J(\theta) = \mathbb{E}_{\mathbf{x} \sim \rho, \mathbf{y} \sim \pi_\theta}\left[r_\psi(\mathbf{x}, \mathbf{y})\right] - \beta \mathbb{KL}(\pi_\theta \| \pi_{\text{ref}}), \qquad (6)$$

and optimize our policy $\pi_\theta$ against that reward model. The objective requires maximizing a reward function based on a distribution induced by the policy $\pi_\theta$. Thus, evaluating this expected value requires sampling from our policy. We use policy gradient to backpropagate through random samples from our policy.

The objective in Eq. 6 is implemented with the Bradley-Terry model, thus is prone to overfitting. Accordingly, we want to regularize the reward model to avoid $r_\psi(\mathbf{x}, \mathbf{y}) \to \infty$ when $p^\star(\mathbf{y} \succ \mathbf{y}' \mid \mathbf{x}) = \{0, 1\}$. As mentioned in Sec. 4.1, when $r_\psi(\mathbf{x}, \mathbf{y}) \to \infty$, the KL regularization $\beta$ vanishes. In particular, we find that the scarce dataset in text-to-motion generation leads to overfitting: the reward model's training loss converges to $0$ while the validation loss increases.

During policy optimization, due to the overfitted reward, the policy tries to hack the reward function and selects tokens that are very improbable under the reference model. As a result, the KL divergence explodes, and so does the reward. Accordingly, the value network is also affected by these sudden spikes. We experimentally find it hard to prevent these spikes. In particular, we found that removing dropout is essential to diminishing the spikes. Surprisingly, we observe that preventing these spikes is not related to better performance as evaluated by FID and R-precision. Overall, we find RLHF particularly difficult to tune in our setup, owing to the instabilities resulting from a reward model's inability to evaluate samples accurately. Instead, we recommend using DPO, which we present next.

### 4.3. Direct Preference Optimization

In DPO, we skip the step of learning a reward model and directly train our policy on the preference data. In particular, we rewrite Eq. 4 the reward function as a function of the optimal policy $\pi_{\theta^\star}$ to Eq. 1:

$$r(\mathbf{x}, \mathbf{y}) = \beta \log \frac{\pi_{\theta^\star}(\mathbf{y} \mid \mathbf{x})}{\pi_{\text{ref}}(\mathbf{y} \mid \mathbf{x})} + \beta \log Z(\mathbf{x}). \qquad (7)$$

Originally in RLHF, we had a loss function on the reward functions to turn our preference data into a reward function. We use Eq. 7 to turn the loss function over reward functions in Eq. 5 into a loss function on policies. In particular, we write Eq. 7 in terms of our current policy $\pi_\theta$ instead of the optimal policy $\pi_{\theta^\star}$, which we denote as $\hat{r}_\theta(\mathbf{x}, \mathbf{y})$:

$$\hat{r}_\theta(\mathbf{x}, \mathbf{y}) = \beta \log \frac{\pi_\theta(\mathbf{y} \mid \mathbf{x})}{\pi_{\text{ref}}(\mathbf{y} \mid \mathbf{x})} + \beta \log Z(\mathbf{x}). \qquad (8)$$

Intuitively, the logarithmic ratio yields a positive value when the policy assigns a higher probability to the response compared to the reference model, indicating a preference. Conversely, it results in a negative value when the policy deems the response less probable than what the reference model suggests, signifying a lesser preference. Then, the objective for DPO is:

$$-\mathbb{E}_{(\mathbf{x}, \mathbf{y}_w, \mathbf{y}_l) \sim \mathcal{D}}\left[\log \sigma\left(\hat{r}_\theta(\mathbf{x}, \mathbf{y}_w) - \hat{r}_\theta(\mathbf{x}, \mathbf{y}_l)\right)\right]. \qquad (9)$$

| | Alignment | | | | Quality | | |
|---|---|---|---|---|---|---|---|
| Method | Top-1↑ | Top-2↑ | Top-3↑ | MM Dist↓ | MModality↑ | FID↓ | Diversity→ |
| Real motion | 0.494±0.002 | 0.677±0.002 | 0.769±0.002 | 3.224±0.008 | - | 0.002±0.000 | 9.463±0.073 |
| MotionGPT [19] | 0.405±0.002 | 0.567±0.002 | 0.658±0.002 | 4.027±0.011 | **3.495±0.162** | **0.178±0.008** | **9.393±0.086** |
| RLHF [26] | 0.415±0.002 | 0.581±0.003 | 0.673±0.002 | 3.908±0.016 | 3.196±0.123 | 0.217±0.009 | 9.303±0.089 |
| DPO [28] | **0.426±0.002** | **0.595±0.002** | **0.689±0.002** | **3.782±0.014** | 2.523±0.091 | 0.219±0.007 | 9.356±0.077 |

Table 1. **Preference data improves alignment.** We find that DPO performs better than RLHF. It is important to note that the FID metric is an inaccurate measure of the quality of the motion. In particular, our labelers prefer outputs from DPO over MotionGPT.

We directly train our policy with Eq. 9 on the preference dataset. Its gradient formula yields a very intuitive understanding of the optimized objective: it increases the likelihood of the preferred sample and decreases the likelihood of dispreferred samples. In formulae, each gradient step is:

$$-\beta\mathbb{E}_{\mathcal{D}}\left[w(\mathbf{x},\mathbf{y}_w,\mathbf{y}_l)\left[\nabla_\theta\log\pi(\mathbf{y}_w\mid\mathbf{x})-\nabla_\theta\log\pi(\mathbf{y}_l\mid\mathbf{x})\right]\right)\right],$$
$$(10)$$

where

$$w(\mathbf{x},\mathbf{y}_w,\mathbf{y}_l)=\sigma(\hat{r}_\theta(\mathbf{x},\mathbf{y}_l)-\hat{r}_\theta(\mathbf{x},\mathbf{y}_w)),\quad(11)$$

is a per-sample weight [45, 46] that gives a higher weight when the reward model is wrong.

However, it is important to remember that DPO is still prone to overfitting as it also relies on the Bradley-Terry model. Moreover, DPO is limited to two points within the data, optimizing to maximize one while minimizing the other. In contrast, RLHF provides a training signal where the reward model generalizes via trial and error, thereby acquiring more information. We alleviate overfitting with a variant of DPO: Identity Preference Optimization (IPO) [3]. Specifically, IPO does not rely on the Bradly-Terry model by setting Ψ as the identity function.

## 5. Experiments

We organize this section as follows. First, we present details of our implementation of both methods: RLHF and DPO. Second, we present the evaluation metrics that follow standard practice in text-to-motion generation. Then, our main results show an improvement in alignment with text, compared with MotionGPT. Finally, we present ablations to understand key design choices in DPO. In particular, we find that proper regularization is important in DPO.

During all training runs, we train for 20 epochs in total and take the epoch with the best validation set performance on HumanML3D [14]. We evaluate quantitatively on the HumanML3D test set and qualitatively on our human preference test set.

**Implementation Details.** Our implementations of RLHF and DPO build upon TRL [39]. We implemented RLHF with separate value and policy networks because empirically, we observed greater training stability. The value

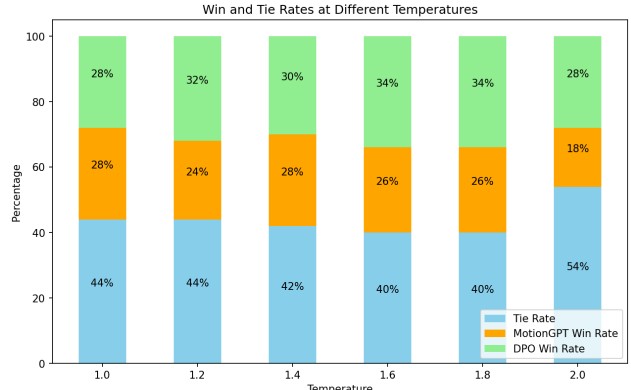

Figure 3. **Humans prefer DPO outputs over outputs from MotionGPT.** MotionGPT trained on motion data with DPO (in green) has a higher win rate. The win rate is computed on prompts never seen by the model.

network is initialized to the reward model with an additional scalar head that predicts a scalar per token (initialized with Gaussian mean 0.0 and standard deviation 0.2, bias initialized to 0.0). The policy network is initialized to the finetuned MotionGPT checkpoint[1]. We remove all dropouts in the value and policy models because when dropouts are present, they cause the KL reward to be stochastic since the SFT model is stochastic. For better performance, we add reward margin (3 for "Much better," 2 for "Better," 1 for "Slightly better") [36], reward whitening, and score scaling [53]. We performed a hyperparameter sweep and found that the best hyperparameters are batch size 32, learning rate 1e-5, NEFTune noise alpha 0.1 [18], fixed KL with no adaptive KL controllers, and initial KL coefficient 0.05.

For the DPO model, we initialize it to the finetuned MotionGPT checkpoint[1]. We performed a hyperparameter sweep and found that the best hyperparameters are batch size 64, learning rate 1e-3, no label smoothing, PEFT [22] with LoRA [16] (rank 8, alpha 16, dropout 0.05), Beta 0.1, no dropouts in the model, and IPO loss [4].

**Evaluation.** We categorize popular metrics [14] in text-to-motion generation into alignment and quality. In particular, alignment is related to the alignment of the text

---

[1]https://huggingface.co/OpenMotionLab/MotionGPT-base

| Percent Data | Alignment | | | | Quality | | |
|---|---|---|---|---|---|---|---|
| | Top-1↑ | Top-2↑ | Top-3↑ | MM Dist↓ | MModality↑ | FID↓ | Diversity→ |
| 100% | **0.426±0.002** | **0.595±0.002** | **0.689±0.002** | 3.782±0.014 | 2.523±0.091 | 0.219±0.007 | 9.356±0.077 |
| 80% | 0.421±0.002 | 0.590±0.002 | 0.682±0.003 | 3.835±0.014 | **2.760±0.118** | **0.204±0.007** | **9.368±0.059** |
| 60% | 0.417±0.002 | 0.585±0.002 | 0.677±0.002 | 3.872±0.011 | 2.594±0.100 | 0.233±0.008 | 9.334±0.070 |
| 40% | 0.420±0.002 | 0.587±0.002 | 0.680±0.002 | 3.845±0.012 | 2.731±0.104 | 0.212±0.006 | 9.340±0.072 |
| 20% | 0.422±0.002 | 0.591±0.002 | 0.685±0.002 | **3.775±0.015** | 2.236±0.086 | 0.252±0.007 | 9.356±0.068 |

Table 2. **More preference data helps.** Our analysis reveals that an increased volume of preference data enhances performance in both alignment and quality metrics, although the impact diminishes with more data. Our results demonstrate that DPO does not need a significant amount of data to exhibit performance gains.

| Loss Type | Alignment | | | | Quality | | |
|---|---|---|---|---|---|---|---|
| | Top-1↑ | Top-2↑ | Top-3↑ | MM Dist↓ | MModality↑ | FID↓ | Diversity→ |
| IPO [4] | **0.426±0.002** | **0.595±0.002** | **0.689±0.002** | **3.782±0.014** | 2.523±0.091 | **0.219±0.007** | 9.356±0.077 |
| KTO [10] | 0.416±0.003 | 0.585±0.002 | 0.678±0.002 | 3.867±0.011 | **3.099±0.104** | 0.241±0.008 | 9.315±0.068 |
| Hinge [28] | 0.418±0.002 | 0.588±0.002 | 0.682±0.003 | 3.828±0.010 | 2.843±0.116 | 0.252±0.008 | **9.362±0.052** |
| Sigmoid [28] | 0.418±0.003 | 0.586±0.002 | 0.679±0.003 | 3.847±0.012 | 2.831±0.100 | 0.254±0.008 | 9.354±0.076 |

Table 3. **IPO loss performs best.** The IPO [3] variant of DPO is designed to alleviate overfitting due to the Bradley-Terry model.

prompt with the generated motion. In contrast, quality is independent of the text prompt and measures the quality of the motion. R-Precision evaluates motion-to-text retrieval accuracy based on Euclidean distances between motion sequences and text descriptions, reporting Top-1, Top-2, and Top-3 accuracies. FID measures the distribution disparity between generated and real motion using extracted motion features. MM-Dist calculates average Euclidean distances between text and generated motion features. Diversity analyzes motion variety via average Euclidean distances among randomly sampled pairs of motion. MModality generates multiple motion sequences per text description, forms pairs, and computes their average Euclidean distances.

| Metrics | w/ PEFT | w/o PEFT |
|---|---|---|
| Top-1↑ | **0.426±0.002** | 0.394±0.001 |
| Top-2↑ | **0.595±0.002** | 0.555±0.002 |
| Top-3↑ | **0.689±0.002** | 0.646±0.002 |
| MM Dist↓ | **3.782±0.014** | 4.097±0.016 |
| MModality↑ | 2.523±0.091 | **3.285±0.114** |
| FID↓ | **0.219±0.007** | 0.276±0.006 |
| Diversity→ | **9.356±0.077** | 9.266±0.063 |

Table 4. **PEFT is an important component for preference learning.** We find that PEFT significantly contributes to the success of DPO by regularizing the model's training.

**Main results.** In Tab. 1, we show our RLHF and DPO results compared to the reproduced MotionGPT baseline[2]. Our results reveal that both RLHF and DPO outperform MotionGPT across all alignment metrics, suggesting greater alignment with text compared to the baseline. Furthermore, when considering quality metrics, RLHF and

[2]https://github.com/OpenMotionLab/MotionGPT/tree/main

DPO demonstrate comparable performance to MotionGPT, suggesting their efficacy in producing high-quality outputs. Notably, our findings highlight DPO's superiority over RLHF in alignment metrics, underscoring its potential as a more effective approach for learning from preferences in motion generation tasks. Since the FID metric is not an accurate measure of the actual quality of the motion, we perform human evaluation of MotionGPT generation against DPO. We compare MotionGPT baseline generations and DPO generations at different temperatures. Given 50 random prompts taken from the test set of our preference dataset, we ask labelers to pick which generation is the best or mark a tie if they cannot make a choice. In Fig. 3, we show the DPO win rate, MotionGPT win rate, and tie rate. On average, DPO generations perform better than MotionGPT generations at all temperature levels, indicating not only that humans prefer DPO outputs over MotionGPT outputs, but also that its good performance is sustained at different temperature levels.

**Ablation.** As we have seen in Sec. 4, an important aspect of preference learning is the trade-off between optimizing the reward model and the KL regularization. Additionally, since DPO does not suffer from reward hacking and performs better than RLHF, we perform our ablation studies on our DPO baseline. First, we try different parameters that directly or indirectly improve the regularization of the reference model during training. In Fig. 5, we find that a $\beta$ around 0.10 performs the best; overall, the model is robust to different choices of $\beta$, underscoring the robustness of DPO. Second, we find that IPO [3] performs best compared to other variants of the DPO loss (where sigmoid is the standard Bradley-Terry model used in the original DPO method). As mentioned in Sec. 4, IPO was specifically

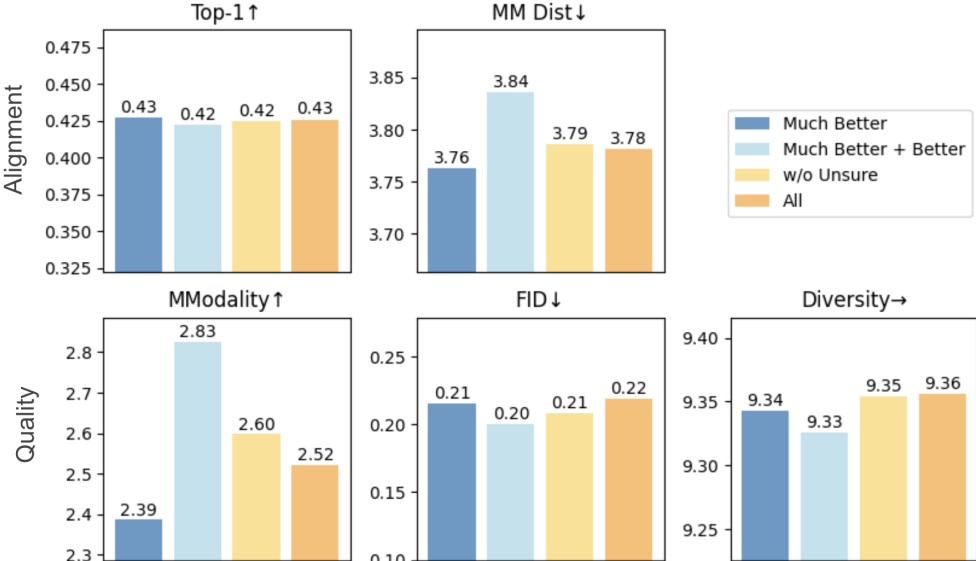

Figure 4. **Samples with preference degrees "Much better" and "Better" provide most of the performance gains.** Adding in "Slightly better" and "Negligibly better/unsure" samples slightly improves alignment but decreases quality.

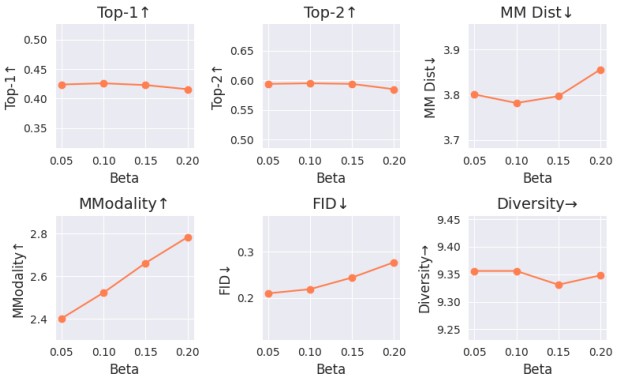

Figure 5. **Model is robust to choices of $\beta$.** Values of $\beta$ increasing from 0.05 to 0.20 generally do not impact alignment.

designed to alleviate overfitting due to the Bradley-Terry model. Third, while LoRA was designed to reduce the cost of training, we observe that it plays an important role in regularizing the model. Tab. 4 shows significant gains from using LoRA. Finally, we study how the scale of the dataset affects training, both in terms of the quantity and the degree of preference. Tab. 2 shows that more data helps. However, we find the gains are not significant, showing that current text-to-motion generative methods do not require a significant amount of preference data to observe improvements. Additionally, in Fig. 4, we train our models on the different preference splits. We find that samples labeled as "Much better" provide most of the performance gains. Our results suggest that labelers should focus on labeling samples with

a considerable visual difference.

# 6. Discussion

This paper is the first work that explores preference learning for text-to-human generation, *i.e.*, a cheaper supervision from human labelers for text-to-human generation. By annotating 3,528 preference pairs and introducing a degree of preference for each choice, we have laid the groundwork for more nuanced and human-like motion generation capabilities. Our pioneering efforts have shown that labelers significantly favor the outputs generated by MotionGPT when it is trained with preference data, highlighting the potential of preference learning in enhancing the alignment of generated motions across various settings.

This paper is limited to exploring preference data on MotionGPT. It would be valuable to analyze the transferability of such a dataset to other models. Additionally, we did not use the skipped samples as both samples did not generate perceptually realistic motion, while the unsure samples generated at least realistic motion. It would be interesting to see how these samples can be leveraged, for instance, with unlikelihood learning on these samples. One can also extend the annotation at the temporal or spatial level for fine-grained supervision. Prior work in image generation used a reward model trained on preference data as a metric for evaluating generation. Similarly, it would be a valuable metric for text-to-motion generation as R-precision and FID correlate poorly with human evaluation. Moreover, it would interesting to study preference learning on bigger datasets such as Motion-X [21].

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
