# OpenReview forum: "Exploring Text-to-Motion Generation with Human Preference"
_thecvf.com/CVPR/2024/Workshop/HuMoGen — CVPR 2024 Workshop HuMoGen Submission_

### Official Review · Reviewer_sU1r · 2024-03-31
**A Valuable Work Requiring Clearer Exposition**

**Rating:** 4
**Confidence:** 3

**Review:**

While the overall idea of the article might not be groundbreaking, it indeed validates important issues within motion generation and offers valuable insights for future research. I think the main issue with the article is its somewhat confusing organizational structure, coupled with a lack of intuitive visualizations of the results.
Here are some minor issues:
- The authors frequently claim that their method surpasses labeling motions because it obviates the need for motion capture devices. However, such claims can be somewhat misleading, as their approach also necessitates a model that has been pre-trained on paired data. Moreover, the mocap and labeling processes are often not conducted simultaneously, and there is an abundance of mocap data that has not yet been labeled. Therefore, I believe it would be more accurate to frame the comparison as between collecting "more" labeled data and gathering some preferred data.
- 015-017: "greatly improve", however, the subsequent metrics did not show significant improvement, and the authors also observed that labeling more preference data did not yield more benefits.
- 024-028: "current motion generation research", I think authors mean text-to-motion generation research.
- LINE 292 (EQU4): Z(x) is not defined here, which should be some integration over x
- LINE 296-299: It feels somewhat odd to discuss this topic here. The author mentions the issue of overfitting and then moves on to a completely unrelated topic in the following paragraph. I initially expected the next section to offer a solution to overfitting. If the author does not intend to address the issue here, perhaps it would be more appropriate to bring it up before the IPO section?
- LINE 331-338: same as above
- Tab 4 and LINE 490: In table, authors compare DPO with/without PEFT, while in LINE 490, they say LoRA is important. While there may be significant overlap in implementation or content, it's important to maintain consistency in how it's described.
- I think incorporating pseudocode would greatly enhance the clarity of the article's presentation.

Here are some major issues:
- The authors have shown that after RLHF or DPO, there is a decrease in the quality metrics. Have they considered proposing alternative metrics? It's noteworthy that the quality metrics mentioned in the article are mostly based on statistical indicators, fundamentally assessing whether the distribution of generated motions and the ground truth are statistically similar. However, the authors have fine-tuned using additional preference data, which could cause a certain shift in the distribution of generated motions. Is it possible to define metrics that are not based on overall statistical data but instead on the evaluation of individual motions?
- I believe the authors should showcase a few motions that are performed better after DPO in the article, providing an intuitive comparison with MotionGPT.

---

### Official Review · Reviewer_syZB · 2024-04-01
**The authors point out the explorations using human preference data to improve t2m generation both in quality and text alignment manner. The authors argues that as currently the field of t2m relays of sכ mall datasets collected with expensive motion capture systems and annotators, it can be easier generating high quality training data by getting experienced people (as annotators) to compare two generate motions to decide which is better aligned to the text. They created a new dataset of human preferences for motion generation and show that using this data can improve the results of t2m quality and alignment. By that the authors are pioneering the RLHF and human preferences in the motion domain.**

**Rating:** 4
**Confidence:** 4

**Review:**

The authors point out the explorations using human preference data to improve t2m generation both in quality and text alignment manner. The authors argues that as currently the field of t2m relays of sכ mall datasets collected with expensive motion capture systems and annotators, it can be easier generating high quality training data by getting experienced people (as annotators) to compare two generate motions to decide which is better aligned to the text. They created a new dataset of human preferences for motion generation and show that using this data can improve the results of t2m quality and alignment. By that the authors are pioneering the RLHF and human preferences in the motion domain.


Pros
- Pioneering in the field of T2M RLHF field - generated a dataset to tackle this issue
- Although not innovative as its self, the authors bring the field of RLHF to the human motion.
- The authors collected and release an important dataset that can be later used for additional work.


Cons
- From the way that the sup. As provided, it is extremely hard to understand the improvement claimed in the paper.
- As in contrast to what has been done in T2I domain, the data generated only by one specific model instead of generating a more diverse dataset.
- The authors did not provide a visual comparison figure to prove that the RLHF/DPO is better than the baseline model.
- The additional work doesn't show significant improvement in quality over the baseline (as can be understood from the user study).


Questions and suggestions for the next revision:

While the labelled data of comparison between two motions includes also the “how much better the motion is”, this signal was not used during the training. Can you explain why you are not using it, and suggest a way to use this signal as well?
Please add qualitative comparison between the methods and the baseline.
It is not clear to me how the user study was conducted, how many pairs were generated, and how many users conducted the user study ?

Overall, the paper tackles an important and rising task that was already adopted by the t2i community, bringing it and emphasising the importance of collecting human preferences in the t2m field. Although I couldn’t find any significant innovation within the method, I do think that this is an important work and thus I would recommend accepting the paper.

---

### Meta-Review · Area_Chair_ytHQ · 2024-04-05

**Recommendation:** Accept

**Metareview:**

The paper received positive reviews. The reviewers appreciated the idea of introducing RLHF and DPO in the motion synthesis context. However, a recurring concern is regarding the quality of the results and whether the proposed modules actually help in the generation. Nonetheless, the paper is a step in an interesting direction and would be of interest to the motion generation community. The authors are encouraged to augment the supplementary material with more examples of the failure/success cases of the method for the camera ready.

---

### Decision · Program_Chairs · 2024-04-06

**Decision:**

Accept

**Comment:**

The paper will be published as part of the official CVPR workshop proceedings upon submission of the camera-ready version.